

**Organic Matter Database (OMD): Consolidating global residue data from**
**agriculture, fisheries, forestry and related industries**
**Gudeta Weldesemayat Sileshi[1,2]\*, Edmundo Barrios[1], Johannes Lehmann[3,4], Francesco N.**
**Tubiello[5]**
[1]Plant Production and Protection Division (NSP), Food and Agriculture Organization of the United Nations (FAO);
[2]Department of Plant Biology and Biodiversity Management, Addis Ababa University, Addis Ababa, Ethiopia
[3]Soil and Crop Sciences, School of Integrative Plant Science, Cornell University, Ithaca, NY, USA.
[4]Cornell Atkinson Center for Sustainability, Cornell University, Ithaca, NY, USA.
[5]Agri-Environmental Statistics, Food and Agriculture Organization of the United Nations (FAO).
***Corresponding author**: Gudeta Weldesemayat Sileshi  e-mail: sileshigw@gmail.com



## Abstract

Agricultural, fisheries, forestry and agro-processing activities produce large quantities of residues, by-products and waste materials every year. Inefficient use of these resources contributes to greenhouse gas emissions and non-point pollution, imposing significant environmental and economic burdens to society. Since many nations do not keep statistics of these materials, it has not been possible to accurately quantify the amounts produced and potentially available for recycling. Therefore, the objectives of the present work were to provide: (1) definitions, typologies and methods to aid consistent classification, estimation and reporting of the various residues and by-products; (2) a global organic matter database (OMD) of residues and by-products from agriculture, fisheries, forestry and related industries; and (3) preliminary estimates of residues and by-products potentially available for use in a circular bio-economy. To the best of our knowledge, the OMD is the first of its kind consolidating quantities and nutrient concentrations of residues and by-products from agriculture, fisheries, forestry and allied industries globally. The OMD and its associated products will be continuously updated as new production data are published in FAOSTAT, and this information is expected to contribute to evidence-based policies and actions in support of sustainable utilization and the transition towards a circular economy. The estimates in OMD are available only at the national level. Due to the lack of uniform methodology and data across countries, it was difficult to accurately estimate the quantities of all agricultural, fisheries and forestry residue and by-products. Therefore, we strongly recommend investment in the inventory of agricultural, fisheries and forestry residues, by-products and wastes for use in a circular bio-economy and as amendments.

**Keywords:** Agro-processing; anaerobic digestate; biochar; bioeconomy; compost; manure



## 1. Introduction

Agricultural, fisheries, forestry and agro-processing activities produce large quantities of residues,
by-products and waste materials every year (Gontard et al., 2018; Lopes and Ligabue-Braun, 2021;
Millati et al., 2019). A large proportion of the residues and by-products is either burnt or
inappropriately disposed of, without further use (Domingues et al., 2017; FAO, 2022a;
Venkatramanan et al., 2021). Conversely, a significant portion of these residues could enter a
circular bio-economy. This inefficient use of resources limits the achievement of sustainability
goals, contributing for instance to greenhouse gas (GHG) emissions and non-point pollution that
impose significant environmental and economic burdens to society (Gontard et al., 2018). Burning
crop residues is a major contributor to dangerously high levels of air pollution and emission of
greenhouse gases (FAO, 2022a; Oanh et al., 2018; Venkatramanan et al., 2021). In 2019 alone,
around 458 million tonnes of crop residues were burnt globally resulting in 1238 kilo tonnes of
methane ($CH_4$) and 32 kilo tonnes of nitrous oxide ($N_2O$) emissions (FAO, 2022a). Burning
agricultural residue also results in substantial losses of nutrients in the residue. For example,
burning crop residues results in nearly complete loss of the organic carbon and nitrogen, and loss of
25% of the phosphorus, 20% of the potassium and 5–60% of the sulphur (Dobermann and Fairhurst,
2002). In addition to causing air pollution and respiratory ailments in human, burning also removes
opportunities for adding value to crop residues (Lin and Begho, 2022; Oanh et al., 2018;
Venkatramanan et al., 2021). The residues could in principle be better managed to increase soil
fertility and productivity, and to mitigate greenhouse gas emissions (Lu et al., 2014; Liu et al.,

2015).

Over the years, agriculture has increasingly depended on synthetic fertilizers to meet crop

nutrient demands. The increased fertilizer use and inefficient fertilizer management practices have
led to large nutrient losses to the environment in some regions (FAO, 2022b; Singh and Craswell,
2021). On the other hand, farmers in low-income countries have limited access to fertilizer inputs,
and this has led to depletion of native soil nutrient stocks from croplands. Access to fertilizers has





been further limited by the recent war by Russia on Ukraine, which disrupted a large portion of the
global fertilizers supply. The resultant increases in prices are likely to constrain fertilizer use by
farmers into the foreseeable future (FAO, 2022c; Behnassi and El Haiba, 2022). In 2020, the
production and use of synthetic fertilizers resulted in GHG emissions of 1.0 Gt $CO_2$ equivalent, of
which 62% (or 0.63 Gt $CO_2$) is emitted when nitrogen fertilizers are used on croplands (FAO,
2022d; Tubiello et al., 2022).
A growing body of evidence from meta-analyses suggests that the combined use of organic
inputs and inorganic fertilizers can increase fertilizer use efficiency (Ba et al., 2022; Chivenge et al.,
2011; Sileshi et al., 2019; Wang et al., 2020; Zhang et al., 2020; Melo et al., 2022). There is also a
growing consensus that judicious use of agricultural residues can partially substitute for inorganic
fertilizers (Fan et al., 2021; Huang et al. 2013; Zhang et al., 2020) and thereby contribute to
enhancing the sustainability of food production by reducing costs and carbon footprints while
reducing pollution caused by nitrate leaching (Zhang et al., 2020). The savings resulting from
recycling agricultural residues and wastes can also be an important contribution to national and
local economies. Recycling of organic residues, by-products and wastes can also address waste
management problems and reduce GHG emissions from residues and wastes (Andrews et al., 2021;
FAO, 2022a).
Yet, the potential contributions of agricultural, fisheries and forestry residues and by-products
to soil health improvement and carbon management has not been estimated fully. This is largely due
to a lack of country statistics on the production of residues and by-products from agriculture,
fisheries and forestry, which makes it difficult to accurately quantify the amounts produced and
available for recycling. The designation of residues as a resource, by-product or waste may also not
always align with how the material is subsequently managed or its potential utility as a soil
amendment. For example, livestock manure may be classified as a waste in some jurisdictions but
not in others, whether or not it is subsequently used as an organic fertilizer. Importantly, a clear
typology of residues and by-products also does not exist in many regions. This hinders the



systematic documentation and reporting of the different categories of organic resources.
Information is also scant on the quality of most of the residues produced. The quality of organic
resource varies with the plant species, plant parts and their maturity level (Palm et al., 2001; Cobo
et al., 2002), and determination of the quality attributes using traditional laboratory methods is both
timely and costly (Shepherd et al., 2003). Despite these challenges, Palm et al. (2001) published an
organic resource database containing data on plant species and plant part, resource quality,
decomposition rates, N release rates, digestibility indices and site characteristics. Rapid plant
nutrient analysis based on spectroscopic methods have been developed (Shepherd et al., 2003), and
complemented with methods assessing functional differences (e.g., carbon and nitrogen release
rates, digestibility) (Vanlauwe et al., 2005). Additional efforts to make this organic resource data
useful included a decision support system for testing the four different management categories of
organic resources as determined by their nitrogen, lignin, and polyphenol contents (Palm et al.,
2001; Vanlauwe et al., 2005). A related effort is the Phyllis database developed by the Energy
Research Centre of the Netherlands (ECN, 2018) primarily focussing on biomass properties that are
relevant to bioenergy and biochar production. Data on primary crop and animal products are
available through FAOSTAT, but equivalent data for quantities of residues are not available
(Ludemann et al., 2023; Woolf, 2020).

In 2020 the Food and Agriculture Organization (FAO) of the United Nations commissioned a

scoping study to assess the state of organic resource databases in the agriculture sector and related
industries (Woolf, 2020). The study arrived at the following conclusions: (1) large uncertainties
exists in the annual production of crop residues; (2) the fate and use of residues and wastes is poorly
quantified in many regions of the world; (3) existing decision tools and classification schemes for
residue biomass are not well suited for allocating resources amongst a comprehensive portfolio; (4)
data on residue biomass composition and properties are diffuse, have large gaps, and rarely relate
composition to production conditions; and (5) paucity of data on residue biomass production,
composition and fate is a critical constraint on improving resource-use efficiency (Woolf, 2020).
Further, the study recommended the development of a global biomass resource database to support
sustainable development goals. Therefore, a global database providing biomass estimates of the
different residue and by-products is urgently needed for practitioners and policy-makers to quickly
refer to when making decisions. Accordingly, the objectives of the present work were to provide:
(1) definitions, typologies and methods to aid consistent classification, estimation and reporting of
the various residues and by-products; (2) a global organic matter database of residues and by-
products from agriculture, fisheries, forestry and related industries; and (3) preliminary estimates of
residues and by-products potentially available for use in a circular bio-economy. Wherever possible,
this work also tried to highlight the competing uses of the various residues and the challenges and
opportunities for their use as soil amendments.

**2. Methods**
To guide development of the OMD, a review of the literature was performed. This was aimed at
identifying the various categories and a typology (systematic classification) of organic residues and
by-products, their competing uses and the challenges and opportunities for their use as soil
amendments. The review also aimed at identifying industry's best-practices and conversion factors
for estimating agricultural, fisheries and forestry residues and agro-processing by-products.
**2.1. Data used for creating the OMD**
The OMD was designed to provide data on both quantity and quality of residues and by-products.
Residue datasets were estimated from the FAOSTAT and FishStatJ databases. FAOSTAT provides
free access to historical data on food, agriculture, forestry, trade, and land use for over 200 countries
and territories. Data on production of primary crop and animal products were extracted from
FAOSTAT's Crop and Livestock Products database (https://www.fao.org/faostat/en/#data/QCL),
while data on forestry residues came from FAOSTAT's Forestry Production and Trade database
(https://www.fao.org/faostat/en/#data/FO) (FAO, 2023). In the case of capture fisheries and
aquaculture, production quantity (in tonnes live weight) came from FAO's FishStatJ statistical





software (https://www.fao.org/fishery/static/FishStatJ) for the periods 2015–2019 for selected
species in each country/territory.

Not only the quantity, but also the quality of residues, is important for their use in soil

amendment. Therefore, a supplementary database was created consolidating data on the nutrient
concentrations of various residues to complement the OMD. The concentrations of carbon,
macronutrients (nitrogen, phosphorus, potassium), micronutrients (sulphur, calcium, magnesium),
lignin, polyphenols and ratios for crop residues and manure were compiled from existing databases
(e.g., Cornell Substrate Composition Table, FAOSTAT, Phyllis database), International Panel on
Climate Change guidelines (IPCC) default values (IPCC, 2019) and the scientific literature (e.g.,
Ludemann et al., 2023 on crop residues, and Shen et al., 2015; Sileshi et al., 2017 on manure).
Wherever available, the range of values (minimum and maximum) available in OMD and IPCC
default values are summarized in Table 1. All values were reported on dry matter basis. The
moisture contents of most residues have not been reported in the original publications and therefore
values should be used with caution.
**Table 1**. Range of values (minimum and maximum) reported for the carbon, nitrogen (N), phosphorus (P),
potassium (K), calcium (Ca) and magnesium (Mg) concentrations of crop residues and manure (on dry
matter basis). Values were summarized from the OMD supplementary database described above.

| Residue | Carbon (%) | Nitrogen (%) | C:N ratio | P (%) | K (%) | Ca (%) | Mg (%) |
|---|---|---|---|---|---|---|---|
| Barley straw | 47 | 0.9* (0.5-0.7)) | | 0.09-1.03 | 1.11-1.18 | | |
| Coconut shell | 53 | 1.43 | 37 | 0.18 | 0.50 | 0.36 | 0.20 |
| Cocoa beans | | 2.8 | | 0.18 | 0.62 | | |
| Cocoa pod husks | | 0.75 | | 0.23 | 1.02 | | |
| Groundnut straw | 42 | 1.30 | 30 | 0.15-0.20 | 1.31-2.19 | 1.97 | 1.15 |
| Groundnut hull | 49 | 1.2-2.16 | 28 | 0.37 | 1.27 | 1.96 | 0.77 |
| Maize stalks | 55 | 0.81-1.26 | 69 | 0.15-0.37 | 1.20-1.61 | 0.35 | 0.48 |
| Rice straw | 45-61 | 0.64-1.69 | 78-88 | 0.05-0.11 | 1.16-2.10 | 0.42-1.2 | 0.3-0.52 |
| Rice husk (hull) | 39-52 | 0.48-0.70 | 70-106 | 0.11-0.46 | 0.28-1.3 | 0.21-0.34 | 0.09-0.40 |
| Rice bran | 50-55 | 2.0-2.4 | 18-22 | 3.60-4.47 | 1.43-2.45 | 0.13-0.35 | 1.11-1.78 |
| Sorghum stalks | 53 | 0.7* (0.7-1.4) | 73 | 0.18-0.25 | 1.50-1.94 | 0.60 | 0.62 |
| Soybean straw | 51 | 0.8* (1.73-2.0) | 40 | 0.14-0.19 | 0.97-1.63 | 0.18 | 0.15 |
| Wheat straw | 47-55 | 0.7* (0.3-1.4) | | 0.07 | 0.86-0.92 | | |
| Manure – dairy cows | 4.3-61 | 2.9* (0.3-4.0) | 16* (1-98.8) | 0.01-3.2 | 0.03-6.1 | 0.02-3.5 | 0.01-1.9 |
| Manure – non-dairy | | 2.3* | 19* | | | | |
| Manure – swine | 16-47 | 4.1* (0.9-4.4) | 11* (8-26.1) | 0.6-1.8 | 0.9-1.6 | 0.4-1.4 | 0.4-0.8 |
| Manure – poultry | 11-50 | 5.1* (0.5-6.8) | 10* (6-37) | 0.05-3.9 | 0.0-4.7 | 0.02-9.4 | 0.02-4.8 |
| Manure – sheep and goats | 15-49 | 3.3* (0.8-5.1) | 11* | 0.12-0.80 | 0.5-1.8 | 1.1-3.4 | 0.4-1.6 |

* Represents IPCC default values



## 2.2. Definitions and typology


The literature reviewed identified many sources of organic input that can be used for soil
amendment. These include crop residues, agro-processing by-products, forestry and logging
industry residues, manure, poultry and meat processing and fisheries and aquaculture by-products.
Authors have used the terms 'residue', 'by-product', 'co-product', 'waste', when referring to the
various organic resources. Therefore, it was necessary to provide clear definitions and typologies
(systematic classification) to facilitate their consistent estimation and compilation in the OMD. A
clear definition could only be found in relation to an existing EU directive (European Parliament
and Council, 2008; 2008/98/EC), which was adopted herein. Accordingly, a "by-product" is defined
as a substance or object whose primary aim is not the production of that item, whereas "waste" is
defined as any substance or object which the holder discards, intends to discard, or is required to
discard. According to the Directive, an object or substance should be regarded as a by-product only
when certain conditions are met as specified under Article 5. In this paper, this norm was followed
and the term "by-product" was consistently used to refer to side products originating from the food
manufacturing stage. By-products may be products of either primary or secondary processing of
crops, which are available at breweries, wineries, milling and refining facilities (Lopes and
Ligabue-Braun, 2021). Wastes were not included in the OMD as they consist of a wide variety of
materials that may be required to be disposed of in accordance with local legislation. Crop residues,
agro-processing by-products, manure and forestry residues were included in the OMD.
## 2.2.1. Crop residues
Crop residues were defined as plant parts left on the field after harvest including straw of cereals,
pods and stems of legumes, tops, stalks, leaves, and shoots of tuber crops, oil crops, sugar crops,
and vegetable crops, and pruning and litter of fruit and nut trees.





2.2.2. Agro-processing by-products
Agro-processing by-products were defined as products from the food and agriculture industry
(Lopes and Ligabue-Braun, 2021). According to literature reviewed, the main sources of agro-
processing by-products are cereal processing, sugar processing, breweries, the beverage industry,
oil presses and bioenergy production, slaughterhouse by-products and fish processing by-products,
which are further defined below.

*2.2.2.1. Cereal processing by-products*
Cereal processing by-products are defined here as the by-product of rice milling and the multi-stage
process of flour production from cereals such as wheat, rice and maize. In the milling process of
rice, the husk (hull), which is the outer cover of the grain, is removed. Further milling removes the
bran resulting in white rice. Rice husk constitutes about 20% of the dry weight of rice grains (Singh,
2018). The bran is part of cereal grain that could be used in a further milling process or as a
biorefinery feedstock (Caldeira et al., 2020).

*2.2.2.2. Sugar industry by-products*
The by-products from the sugar industry include bagasse (the fibrous residue remaining after the
milling of cane stalks), sugar beet pulp, molasses, and filter press mud, which are available at the
milling and refining facilities.

*2.2.2.3. Brewery and winery by-products*
Spent grain and grape pomace are the main by-product from the brewery and winery industry,
respectively. Barley grain is the main raw material for beer, and ∼20 kg of wet brewer's spent
grains is produced per 100 litres of beer produced (Gonzalez-Garcia et al., 2018). Approximately
75% of grapes produced is intended for wine production, out of which 20–30% represents a by-
product called grape pomace consisting of the skin, pulp, seeds, and stalks (Antonić et al., 2020).




*2.2.2.4. Beverage industry by-products*
The beverage industry manufactures ready-to-drink products such as fruit juice, cocoa, coffee and
tea-based products, soft drinks, energy drinks, milk products, nutritional beverages. The by-
products of fruit processing include the peels, skin, rind and seeds. The main by-products of cocoa
processing are cocoa pod husk, cocoa bean shells and cocoa mucilage. In the initial stage of cocoa
processing, 70–80% of the fruit is discarded and, approximately ten tonnes of shells are generated
for each tonne of cocoa (Dutra et al., 2023).
In making the coffee beverage, approximately 90% w/w dry matter of the coffee cherry is
discarded in the form of husks, parchments, mucilage, silver skin and spent coffee grounds
(Iriondo-DeHond et al., 2020). On wet weight basis, in 100 kg of mature coffee cherries, 39 kg
corresponds to the skin and pulp and 22 kg of mucilage and about 39 kg of parchment is generated
(Iriondo-DeHond et al., 2020).

*2.2.2.5. Oil processing by-products*
The main oil crops include oil palm, coconut, groundnut, soybeans and olives. By-products from
palm oil mills include empty fruit bunches (EFB), palm oil mill effluent, decanter cake, seed shells
and the fibre from the mesocarp. A hectare of oil palm produces 10–35 tonnes of fresh fruit bunch
(FFB) per year on wet weight basis. EFB, fibber, shells and decanter cake account for 30, 6, 3 and
29% of the fresh fruit bunch (FFB), respectively (Embrandiri et al., 2012). EFB is the residue left
after the processing of fresh fruit bunch at the mill. Palm press fibre (PPF) or mesocarp fibre is
produced after pressing fruit or mesocarp to obtain oil. On average, for every tonne of FFB
processed, 120 kg of fibre is produced on wet-weight basis (Embrandiri et al., 2012). Palm kernel
shell (PKS) is difficult to decompose and it has been used as mulch. Decanter cake is another waste
product used as either fertilizer or animal food. Palm oil mill effluent is the outcome of oil





extraction, washing and cleaning processes in the mills. On wet weight basis, about 3 tonnes of oil
mill effluent is produced for every tonne of oil extracted in an oil mill.

Coconuts consists of husks (33–35%), shell (12–15%) and copra (28–30%) on wet weight

basis. According to Onwudike (1996) bout 2,220 kg of dry husks and 1,040 kg of dry shells become
available per hectare per year. Lim (1986) gives figures of 5,280 kg of dry husks and 2,510 kg of
dry shells per ha per year in large-scale estates. Copra production ranges from 0.5–1 tonnes per ha
per year with traditional harvesting on small holdings to 3–9 tonnes per ha for improved clonal
varieties and intensive management (Lim, 1986).

The processing of groundnut oil produces a large portion of peanut meal as a by-product, and

skins and hulls. On wet-weight basis, a 1000 kg of peanuts can generate about 500–700 kg of
peanut meal depending on the procedure of oil extraction (Zhao et al, 2012). An estimated 35–45 g
of skin and 230–300 g of hulls are generated per kg of shelled groundnut kernel (Zhao et al, 2012).
Soybean curd residue is the main by-product of soybean products, and about 1.1 kg of fresh curd
residue is produced from every kilogram of soybeans processed into soymilk or tofu (Khare et al.,
1995). The manufacturing process of the olive oil yields a semi-solid waste called olive cake (30%)
and aqueous liquor (50%). About 10 g of olive cake is produced per kilogram of virgin olive oil
processed (Masella et al., 2014).

*2.2.2.6. Bioenergy by-products*
The main routes in the production of bioenergy are pyrolysis and gasification and anaerobic
digestion (Hamelin et al., 2019; Masoumi et al., 2021). The main bioenergy by-products with
potential use in soil amendment include (1) biochar from thermochemical conversion with pyrolysis
producing bio-oil and gasification producing syngas as the main product; (2) hydrochar from
hydrothermal liquefaction with bio-oil as the main product; (3) digestate from anaerobic digestion
with biogas as the main product; and (4) molasses from lignocellulosic ethanol production with
bioethanol as the main product (Hamelin et al., 2019; Karan and Hamelin, 2021; Masoumi et al.,



2021). Conversion of agricultural residues and by-products into biochar provides an option for
better waste management and reducing the residue volume to be applied (Alkharabsheh et al.,
2021). Biological methods such as digestion and composting do not reliably get rid of contaminants
such as antibiotics, heavy metals and pathogens from agricultural and fisheries residues. Processing
these materials into biochar, however, can destroy pathogens and pollutants such as hormones and
antibiotics given the high temperatures during pyrolysis. In addition, biochar has been to control
plant diseases (de Medeiros et al., 2021; Poveda et al., 2021).

Due to the need for drying the feedstock for pyrolysis that can be energy-intensive and costly

for very wet feedstock, hydrothermal carbonization is considered as an alternative to pyrolysis.
Hydrothermal carbonization is carried out at relatively lower temperatures of 80-240 °C, under
subcritical water pressure (Padhye et al., 2022). The solid output of this process is called hydrochar
(Masoumi et al., 2021; Padhye et al., 2022).

Biogas production involves anaerobic digestion of organic wastes to produce methane (Akbar

et al., 2021; Ma et al., 2022). This process produces large quantities of digestate that can be used as
soil amendment. Since anaerobic digestion deactivates pathogens (Ma et al., 2022), it is also safer
than direct application biowaste. Due to increasing numbers of livestock feeding operations and the
consequent increase in the number of large-scale biogas plants, huge quantities of digestate are
produced in some regions. Digestate probably has more than 80% moisture, whereas hydrochar can
have 20-50% moisture content.

*2.2.2.7. Slaughterhouse by-products*
These consist of poultry and meat processing by-products. Depending on the species, on wet weigh
basis about 20% of meat processing by-products are inedible (Caldeira et al., 2020) and this may be
used for soil amendment.





*2.2.2.8. Fish processing by-products*
Fish processing by-products include the trimmings of fish either in aquaculture or capture fisheries,
for example heads, frames, skin and tails. These materials may constitute up to 70% of fish and
shellfish after processing. Depending on the market, some species are not processed at all, while
others, especially larger fish, are often extensively transformed to fillets or parts of fillets. Fish fillet
yield is species-dependent and is often in the range of 30–50% of the fish on wet weight basis.

2.2.3. Livestock manure
Livestock manure is defined here as the excreta of domestic animals (e.g., poultry, cows, sheep,
horses, rabbits, etc.) including the plant material used as bedding for animals. Two major categories
of manure source are recognized by the IPCC: manure management systems and manure left on
pasture. Manure left on pasture is difficult to collect and therefore largely unavailable for use as soil
amendment. In management systems, manure may be found in liquid (liquid or slurry) or solid form
in cattle, pig and poultry farms. In such systems, cattle produce large quantities of manure, with
dairy cows producing 62 kg per day or about 10% of the weight of an average cow on wet weight
basis (EnviroStats, 2008). Feedlot cattle can generate manure about 5–6% of their body weight each
day or a dry mass of roughly 5.5 kg per animal per day (Font-Palma, 2019). Full-grown milking
cows can produce 7–8% of their body weight as manure per day or roughly 7.3 kg dry mass per
animal per day (Font-Palma, 2019). Bulls, beef cows, steers, heifers and calves produce 42, 37, 26,
24 and 12 kg manure per animal per day, respectively (EnviroStats, 2008). Different categories of
pigs produce 1–4 kg of manure per day, while poultry species produce less than 1 kg of manure per
day.

2.2.4. Forestry residues
Forestry residues can be divided into primary and secondary residues (Karan and Hamelin, 2020).
Primary residues are defined as residues that are left after logging operations (e.g., branches,
stumps, treetops, bark, etc.), whereas secondary residues are by-products and co-products of
industrial wood-processing operations (Karan and Hamelin, 2020). Primary residues were excluded
from the OMD because they are often unavailable for agricultural use. Here, only wood residues
were included. The FAOSTAT definition of wood residues covers wood that has passed through
some form of processing but which also constitutes the raw material of a further process such as for
particle board, fibreboard or energy purposes (FAO, 2022e). This excludes wood chips, made either
directly in the forest from roundwood or made in the wood processing industry (i.e., already
counted as pulpwood or wood chips and particles), and agglomerated products such as logs,
briquettes, pellets or similar forms as well as post-consumer wood.

**2.3. Estimating the quantities produced**
Due to the lack of databases on agricultural residues and by-products, practitioners often use residue
to product ratios (RPR) to estimate residue biomass from data on production of primary products
obtained from local statistics or global databases such as FAOSTAT and EUROSTAT (e.g.,
Bentsen et al., 2014; Bedoić et al., 2019; Karan and Hamelin, 2021; Ronzon and Piotrowski, 2017).
The estimation is sometimes done assuming a mathematical relationship (e.g., linear, logarithmic,
hyperbolic or exponential function) between the primary crop yield and the residue yield (Bentsen
et al., 2014; Ronzon and Piotrowski, 2017). The disadvantage of the RPR is that it is constant over
time and space for a given crop, whereas methods based on mathematical functions can be more
flexible. In this work, the estimation of residues and by-products generally followed IPCC
guidelines (IPCC, 2019) and the FAO guidelines in the Bioenergy and Food Security Rapid
Appraisal user manual for crop and livestock residues (FAO, 2014). In the case of crop residues, the
IPCC provides two alternative methods for estimation of the aboveground crop residue yield
($AG_{DM(T)}$) in kg ha$^{-1}$ on dry mass basis. The first method involves multiplying the harvested crop
yield with the ratio of aboveground dry matter ($R_{AG(T)}$) provided in Table 11.A of IPCC (2019). The
second method involves estimation of residue yields from crop yield using linear equations in Table





11.2 (IPCC, 2019). For any given given crop (T), the two methods are expressed as follows
following the exact IPCC notations:
First method: $AG_{DM(T)} = Crop_{(T)} \times R_{AG(T)}$
Second method: $AG_{DM(T)} = Crop_{(T)} \times Slope_{(T)} + Intercept_{(T)}$
The first method always yields a constant harvest index, most of the times larger than the typical
values reported in the literature (e.g., Ludemann et al., 2023). For example, the IPCC default values
of $R_{AG(T)} = 1$ and 1.2 for maize and barley yield harvest indices of 0.50 and 0.47, while the typical
values are less than 0.47 and 0.41, respectively. As a result, the first method systematically
underestimates residue production relative to the second method. The advantage of the second
method is that it yields a more realistic harvest index commensurate with the grain yield achieved in
a particular country and year. Therefore, the second method was chosen for estimating $AG_{DM(T)}$
from $Crop_{(T)}$ in FAOSTAT for the period 2015-2020. Then, the total annual above-ground residue
production ($AGR_{(T)}$) was calculated for each crop (T) by multiplying $AG_{DM(T)}$ by the harvested area
available in FAOSTAT per country and year for maize, wheat, rice, barley, soybean and groundnut.
The average values of six years (2015–2020) per country were summed across countries to provide
annual aboveground residue production estimates ($AGR_{(T)}$ in tonnes on dry matter basis) for each
region.




**Table 2**. The IPCC equations used for estimation of above-ground crop residue yield ($AG_{DM(T)}$) in tonnes per
ha) from grain yield ($Crop_{(T)}$ in tonnes per ha) from FAOSTAT, and IPCC default values for dry matter
fraction of harvested product and dry matter fraction of aboveground crop residue.

| Crop | IPCC equation for $AG_{DM(T)}$ [†] | IPCC default values | |
| --- | --- | --- | --- |
| | | Dry matter fraction of harvested product, $R_{AG(T)}$ [†] | Dry matter fraction of aboveground crop residue[‡] |
| Wheat | $0.52+1.51*Crop_{(T)}$ | 0.89 | 0.86 |
| Maize | $0.61+1.03*Crop_{(T)}$ | 0.87 | 0.82 |
| Oat | $0.89+0.91*Crop_{(T)}$ | 0.89 | 0.77 |
| Barley | $0.59+0.98*Crop_{(T)}$ | 0.89 | 0.84 |
| Rice | $2.46+0.95*Crop_{(T)}$ | 0.89 | 0.87 |
| Millet | $0.14+1.43*Crop_{(T)}$ | 0.90 | 0.85 |
| Sorghum | $1.33+0.88*Crop_{(T)}$ | 0.89 | 0.85 |
| Rye | $0.88+1.09*Crop_{(T)}$ | 0.88 | 0.85 |
| Groundnuts | $1.54+1.07*Crop_{(T)}$ | 0.94 | 0.90 |
| Dry beans | $0.68+0.36*Crop_{(T)}$ | 0.91 | -- |
| Soybean | $1.35+0.93*Crop_{(T)}$ | 0.91 | 0.85 |

[†]These are all dry matter values at grain moisture contents of 9–13% or dry matter fraction of 0.87–0.91.
[‡] Values are from Ludemann et al. (2023).

Production of agro-processing by-products is often estimated using the RPR and related

coefficients following the FAO guidelines (FAO, 2014). Wherever available, these values defined
as extraction rates, were obtained from FAO's Technical Conversion Factors for Agricultural
Commodities (FAO, 2009). When not available, average values from the literature were used for
estimating the various by-products from the production data in FAOSTAT.

Poultry processing by-products were estimated from the take-off rate, dressed carcass weight

(% of live weight) and stocks (heads) using the following equation:
*Residue = (take-off rate/100)\*average live weight\*(100-% carcass weight)\*stocks*
For each poultry species (chickens, ducks, geese and turkeys) in each country/territory, the take-off
rate (in %), average live weight (kg/animal), and dressed carcass wet weight (in %) were obtained
from FAO's Technical Conversion Factors for Agricultural Commodities (FAO, 2009), while



stocks (number of animals) were obtained from FAOSTAT Crops and livestock products
(https://www.fao.org/faostat/en/#data/QCL).

Similarly, meat processing by-products were estimated from the take-off rate, dressed carcass

weight (% of live weight) and stocks (heads) using the following equation:
*Residue = (take-off rate/100)\*average live weight\*(100-% carcass weight-% hides/skins)\*stocks*
The dressed carcass weight is the weight of the carcass after removal of hide/skin, head, feet, offal,
raw fats, and blood which is often not collected in the course of slaughter. For each species
(buffaloes, cattle, sheep, goats, horses, camels and pigs) in each country/territory, the take-off rate
(in %), average live weight (kg/animal), and dressed carcass wet weight (in %) were obtained from
FAO's Technical Conversion Factors for Agricultural Commodities. As in the poultry species,
stocks were obtained from FAOSTAT Crops and livestock products for each country/territory.
Carcass weight was as defined in FAO's Livestock statistics: Concepts, definitions and
classifications (FAO, 2011).

Residues from capture fisheries and aquaculture species were estimated using the conversion

factors in the Handbook of Fishery Statistical Standards (CWP, 2004) for selected species. In the
fisheries industry, the term "conversion factor" is used principally when converting the volume or
mass (more commonly referred to as the "weight") of a product at one stage to its volume or mass at
another stage in the chain (FAO, 2004). Conversion factors for a particular state of processing vary
according to species and state of processing. The state of processing is hierarchical, and may consist
of the following categories: (a) gutted, (b) headed and gutted, (c) dressed, (d) fillet (skin on or off),
etc. The FAO global inland and marine capture database includes catches for over 2000
species/items (including the "not elsewhere included" categories). Since conversion factors are not
available for all species, first species were ranked based on the number of countries producing and
the total production in 2019. Then the top 6 species were selected for the present analysis because
of availability of conversion factors and the large number of countries involved in their production.
Among the aquaculture species, rainbow trout (*Oncorhynchus mykiss*) was chosen as it was the



topmost grown in aquaculture in 91 countries. In capture fisheries, yellow fin tuna (*Thunnus*
*albacares*), skipjack tuna (*Katsuwonus pelamis*), swordfish (*Xiphias gladius*), Bigeye tuna
(*Thunnus obesus*) and albacore (*Thunnus alalunga*) were chosen for the analysis. Each of these
species were harvested in 96, 90, 83, 79 and 71 countries, respectively. The production quantity was
then converted to residues as follows: Value-(Value/CF) where CF is the indicative factors for
converting product weight to live weight. The FAO database of capture fisheries production covers
only retained catches; data on by-catch (discarded catches) are not included (Garibaldi, 2012). This
means that the by-products can be severely underestimated.
Manure production (in tonnes/year on dry matter basis) was estimated from manure excretion
rate (kg/head/day on dry-weight basis) and stocks (from FAOSTAT) following the FAO guideline
for the different animal categories (FAO, 2014). The general formula for manure production is as
follows:
*Manure production (tonnes/year) = (365\*stocks\*manure excretion rate)/1000*
Since there is no global database which provides country-specific data on manure production, the
FAO tool uses the IPCC default values (FAO, 2014). For each species, average manure excretion
rates were obtained from values compiled from the literature. For the USA, excretion rates were
obtained from ASAE Standards D384.1 of the American Society of Agriculture Engineers (ASAE)
Manure production and characteristics (2005). Manure production was estimated for different
management systems of cattle (non-dairy and dairy) and chicken (broilers and layers) separately
because these are always managed as separate enterprises.
When compiling forestry residues, primary residues were excluded because of the concerns
related to the environmental and economic sustainability of removing them from the forest for soil
application on farm-land. Therefore, the analysis focused on wood residues following the FAO
definition. Data on production quantity of wood residues (item code 1620) in FAOSTAT
([https://www.fao.org/faostat/en/#data/FO)](https://www.fao.org/faostat/en/#data/FO) were used for compiling the OMD. These are reported in
cubic meters solid volume excluding bark on FAOSTAT.



A database of all the coefficients and RPR used in the estimation of the various residues and
by-products is now available in the OMD.

**3. Results**
**3.1. Crop residues**
Maize had the largest global total annual above-ground residue production (~1.28 billion tonnes)
followed by wheat (~1.25 billion tonnes) and rice (~1.11 billion tonnes) (Table 3). The estimated
quantities of crop residue varied widely by continent and region. For example, the largest total
annual production of maize residue was recorded in Northern America including Canada and USA
(~0.41 billion tonnes) followed by Eastern Asia (~0.30 billion tonnes) including China, Democratic
People's Republic of Korea, South Korea and Japan; China accounted for over 99% of the residues
produced in Eastern Asia. The largest wheat residue production was recorded in Southern Asia
(~0.24 billion tonnes) including Afghanistan, Bhutan, India, Iran, Nepal and Pakistan and Sri
Lanka, of which over 67% was produced in India. Rice residue production was highest in Southern
Asia (~0.38 billion tonnes), of which over 70% was produced in India. The global total annual
residue production from soybean was ~0.49 million tonnes, while for groundnuts the corresponding
value was ~0.10 billion tonnes (Table 3). The largest soybean residue production was recorded in
South America (~0.25 billion tonnes) of which Brazil accounted for 61% of soybean residue
production in that region. This was followed by Northern America (~0.16 billion tonnes) of which
USA accounted for 94% of soybean residue production in Northern America.





**Table 3**. Estimated total annual crop residue potentially produced (in 1000 tonnes on dry matter basis) by
selected crops across different regions estimated from FAOSTAT data (see methods).

| | | Maize | Wheat | Rice | Barley | Soybean | Groundnut |
|---|---|---|---|---|---|---|---|
| Africa | Eastern Africa | 42622 | 9530 | 15061 | 2901 | 1534 | 7056 |
| | Middle Africa | 11522 | 30 | 6405 | 0 | 212 | 5782 |
| | Northern Africa | 8534 | 32724 | 5817 | 5676 | 62 | 6279 |
| | Southern Africa | 14502 | 2908 | 8 | 450 | 1995 | 131 |
| | Western Africa | 32457 | 194 | 44747 | 2 | 2614 | 21973 |
| Americas | Caribbean | 929 | 0 | 2279 | 0 | 0 | 99 |
| | Central America | 37518 | 5438 | 1972 | 1105 | 710 | 469 |
| | Northern America | 412953 | 141792 | 11567 | 14628 | 159366 | 5054 |
| | South America | 170584 | 44654 | 34221 | 6342 | 244685 | 2824 |
| Asia | Central Asia | 2485 | 42233 | 1851 | 5727 | 411 | 50 |
| | Eastern Asia | 297844 | 216137 | 302030 | 1731 | 26656 | 25378 |
| | South-Eastern Asia | 53698 | 227 | 293393 | 166 | 2555 | 6118 |
| | Southern Asia | 49564 | 244427 | 383033 | 6744 | 26561 | 16884 |
| | Western Asia | 8152 | 50475 | 1605 | 14488 | 197 | 393 |
| Oceania | Australia and New Zealand | 646 | 39395 | 503 | 12701 | 50 | 23 |
| | Melanesia | 27 | 0 | 21 | 0 | 0 | 11 |
| | Micronesia | 0 | 0 | 0 | 0 | 0 | 0 |
| Europe | Eastern Europe | 86330 | 238524 | 1752 | 45433 | 13930 | 1 |
| | Northern Europe | 156 | 47468 | 0 | 19172 | 0 | 0 |
| | Southern Europe | 25701 | 32708 | 3613 | 12573 | 2617 | 8 |
| | Western Europe | 21935 | 102338 | 115 | 25998 | 912 | 0 |
| | **Total** | **1278157** | **1251201** | **1109994** | **175835** | **485065** | **98533** |


## 3.2. Agro-processing by-products

3.2.1. By-products from processing crops
Globally, maize processing yielded the largest quantity of by-products (~0.12 billion tonnes)
followed by wheat (~0.10 billion tonnes), rice (~0.09 billion tonnes) and barley (~0.04 billion
tonnes) (Table 4). The largest quantity of maize processing by-products was recorded in Northern
America, followed by Eastern Asia and South America. The largest quantity of wheat processing
by-products was recorded in Southern Asia followed by Eastern Europe and Eastern Asia (Table 4).
The global annual production of by-products of coffee, cocoa and oil palm processing were
estimated at 20.5, 5.3 and 170.1 million tonnes (Table 4). The largest quantity of coffee-processing
by-products was recorded in South America, with Brazil producing about 6.5 million tonnes
accounting for over 71% of the annual production in South America. This was followed by South-
Eastern Asia, where Viet Nam produced 3.3 million tonnes annually. The largest quantity of by-
products from cocoa was produced in West Africa, where Cote d'Ivoire accounted for over 60% of



the production in that region. Out of the 170.1 million tonnes of global annual oil palm by-products,
Indonesia accounted for over 59% of the total annual global production.

**Table 4**. Estimated total annual agro-processing by-products of different crops produced (in 1000 tonnes on
dry matter basis) across different regions. All values were estimated using FAOSTAT data (see methods).

| | | Maize | Wheat | Rice | Barley | Soybeans | Groundnut | Coffee | Cocoa | Oil palm |
|---|---|---|---|---|---|---|---|---|---|---|
| Africa | Eastern Africa | 3493 | 727 | 963 | 671 | 63 | 794 | 2051 | 58 | 80 |
| | Middle Africa | 789 | 1 | 219 | 0 | 6 | 824 | 206 | 285 | 2117 |
| | Northern Africa | 827 | 2492 | 569 | 1101 | 4 | 903 | 0 | 0 | 0 |
| | Southern Africa | 1376 | 227 | 0 | 114 | 95 | 20 | 0 | 0 | 0 |
| | Western Africa | 2568 | 14 | 2345 | 0 | 92 | 3227 | 292 | 3295 | 7241 |
| Americas | Caribbean | 68 | 0 | 179 | 0 | 0 | 13 | 110 | 100 | 118 |
| | Central America | 3415 | 439 | 162 | 269 | 32 | 107 | 2244 | 50 | 3130 |
| | Northern America | 41834 | 11050 | 1117 | 3737 | 9525 | 1007 | 5 | 0 | 0 |
| | South America | 16501 | 3440 | 2992 | 1606 | 14294 | 649 | 9145 | 720 | 6096 |
| Asia | Central Asia | 243 | 2983 | 134 | 1214 | 21 | 12 | 0 | 0 | 0 |
| | Eastern Asia | 28988 | 17498 | 27778 | 434 | 1292 | 6338 | 110 | 100 | 118 |
| | South-Eastern Asia | 5073 | 16 | 23094 | 39 | 109 | 1159 | 2244 | 50 | 3130 |
| | Southern Asia | 4484 | 18827 | 29327 | 1517 | 960 | 3007 | 5 | 0 | 0 |
| | Western Asia | 809 | 3860 | 147 | 3362 | 12 | 82 | 9145 | 720 | 6096 |
| Europe | Eastern Europe | 8380 | 18618 | 151 | 11092 | 663 | 0 | 0 | 0 | 0 |
| | Northern Europe | 15 | 3883 | 0 | 5097 | 0 | 0 | 0 | 0 | 0 |
| | Southern Europe | 2565 | 2576 | 333 | 3147 | 155 | 2 | 0 | 0 | 0 |
| | Western Europe | 2196 | 8405 | 10 | 7032 | 50 | 0 | 0 | 0 | 0 |
| Oceania | Australia and New Z | 64 | 2890 | 51 | 2975 | 2 | 6 | 0 | 0 | 0 |
| | Melanesia | 2 | 0 | 1 | 0 | 0 | 2 | 104 | 43 | 1293 |
| | Micronesia | 0 | 0 | 0 | 0 | 0 | 0 | 0 | 0 | 0 |
| | Polynesia | 0 | 0 | 0 | 0 | 0 | 0 | 0 | 0 | 0 |
| | **Total** | **123690** | **97945** | **89569** | **43406** | **27373** | **18149** | **20511** | **5268** | **170137** |


3.2.2. By-products from slaughterhouses
Globally, the largest quantity of residues produced annually was from cattle (16.5 million tonnes)
followed by chicken (10.7 million tonnes) and pigs (6.2 million tonnes), but with wide variation
among regions (Table 5). The largest quantity of by-products from cattle was recorded in South
America (5.31 million tonnes) of which Brazil accounted for 77% of by-products produced in that
region. This was followed by Northern America (4.59 million tonnes of which 94% was in USA)
and Eastern Asia (0.99 million tonnes of which 84% was produced in China). The total annual
production of by-products from chicken processing was largest in North America (6.0 million
tonnes) of which over 99% was produced in the USA. This was followed by East Asia (0.91 million
tonnes) of which China accounted for over 72% of the production in East Asia.






**Table 5**. Estimated total annual quantity of slaughterhouse by-products potentially produced (in 1000 tonnes
on dry matter basis) across different regions. All values were estimated using FAOSTAT data (see methods).

| Continent | UN Region | Cattle | Buffalo | Sheep | Goats | Pigs | Chicken | Turkeys |
|---|---|---|---|---|---|---|---|---|
| Africa | Eastern Africa | 436 | | 84 | 133 | 80 | 65 | 1 |
| | Middle Africa | 141 | | 60 | 2067 | 17 | 12 | |
| | Northern Africa | 162 | 33 | 161 | | 0 | 158 | 7 |
| | Southern Africa | 125 | 723 | 31 | 8 | 11 | 119 | 0 |
| | Western Africa | 306 | | 94 | 168 | 31 | 57 | |
| Americas | Caribbean | 51 | | 2 | 3 | 24 | 79 | 0 |
| | Central America | 450 | | 8 | 5 | 81 | 153 | 1 |
| | Northern America | 4591 | | 25 | 46 | 1072 | 6004 | 51 |
| | South America | 5311 | | 42 | 14 | 272 | 864 | 8 |
| Asia | Central Asia | 321 | | 141 | 12 | 8 | 19 | |
| | Eastern Asia | 994 | 48 | 108 | 117 | 2482 | 906 | 0 |
| | South-Eastern Asia | 206 | 47 | 27 | 41 | 409 | 748 | 0 |
| | Southern Asia | 625 | | 181 | 388 | 39 | 574 | 1 |
| | Western Asia | 175 | 3 | 175 | 45 | 6 | 202 | 6 |
| Europe | Eastern Europe | 433 | 1 | 68 | 5 | 327 | 287 | 38 |
| | Northern Europe | 407 | | 91 | | 303 | 275 | 5 |
| | Southern Europe | 297 | 1 | 66 | 14 | 354 | 14 | 1 |
| | Western Europe | 847 | | 35 | 4 | 671 | 142 | 28 |
| Oceania | Australia and New Zealand | 605 | | 399 | 35 | 35 | 55 | 3 |
| | Melanesia | 2 | | 0 | 0 | 9 | 2 | 0 |
| | Micronesia | | | | | 0 | 0 | |
| | Polynesia | 1 | | 0 | 0 | 1 | 0 | |
| | **Total** | **16487** | **855** | **1797** | **3104** | **6231** | **10735** | **150** |


3.3.3. By-products from fisheries and aquaculture
The estimated annual quantity of by-products potentially produced from processing of selected fish species
in aquaculture and capture fisheries are summarized in Table 6. Among the species grown in aquaculture,
the largest quantity of by-products was produced by rainbow trout (over 0.08 million tonnes) across
91 countries (Table 6). The largest proportion was recorded in Southern Asia (predominantly in Iran
and Tukey), followed by South America (mainly in Peru and Chile) and Northern Europe (mostly in
Norway) (Table 6). Among the capture fisheries species, the largest quantity of by-products was
produced from skipjack tuna harvest (0.14 million tonnes) followed by yellowfin tuna (0.08 million
tonnes).

**Table 6**. Estimated total annual quantity of by-products potentially produced (in tonnes on dry matter basis)
by selected fish species in aquaculture and capture fisheries across different regions. All values were
estimated using FishStatJ data (see methods).

| | | Aquaculture | Capture fisheries | | | | |
|---|---|---|---|---|---|---|---|
| Continent | UN Region | Rainbow trout | Albacore | Bigeye | Skipjack | Swordfish | Yellowfin |
| Africa | Eastern Africa | 90 | 100 | 560 | 4550 | 250 | 3170 |
| | Middle Africa | 0 | 0 | 70 | 360 | 10 | 190 |
| | Northern Africa | 10 | 40 | 20 | 60 | 370 | 10 |
| | Southern Africa | 340 | 210 | 40 | 0 | 100 | 80 |
| | Western Africa | 0 | 40 | 680 | 7100 | 50 | 2980 |
| Americas | Caribbean | 0 | 30 | 320 | 2060 | 10 | 1200 |
| | Central America | 1110 | 20 | 380 | 2150 | 120 | 4580 |
| | Northern America | 2970 | 230 | 210 | 3140 | 230 | 590 |
| | South America | 14150 | 130 | 1880 | 9620 | 1650 | 7110 |
| Asia | Central Asia | 180 | 0 | 0 | 0 | 0 | 0 |
| | Eastern Asia | 5110 | 990 | 960 | 5210 | 260 | 1670 |
| | South-Eastern Asia | 0 | 710 | 2560 | 33290 | 470 | 15580 |
| | Southern Asia | 14730 | 0 | 510 | 13820 | 810 | 13880 |
| | Western Asia | 6860 | 60 | 0 | 130 | 280 | 4920 |
| Europe | Eastern Europe | 6740 | 30 | 0 | 0 | 0 | 0 |
| | Northern Europe | 13150 | 240 | 0 | 10 | 10 | 0 |
| | Southern Europe | 6090 | 300 | 310 | 1680 | 730 | 770 |
| | Western Europe | 5200 | 100 | 100 | 610 | 10 | 790 |
| Oceania | Australia and New Zealand | 0 | 170 | 40 | 180 | 160 | 120 |
| | Melanesia | 10 | 1560 | 1080 | 21470 | 40 | 12920 |
| | Micronesia | 0 | 250 | 1830 | 28450 | 20 | 6250 |
| | Polynesia | 0 | 690 | 240 | 670 | 30 | 540 |
| | **Total** | **76740** | **11790** | **134560** | **5610** | **77350** | **76740** |

### 3.3. Livestock manure
Globally, cattle, buffaloes and chicken produced the largest proportion of the potential annual
manure produced every year (Table 7). Non-dairy cattle produce an estimated 2.23 billion tonnes,
while dairy cattle produce about 0.82 billion tonnes annually on dry matter basis. The largest
quantity of non-dairy cattle manure was produced in South America (where Brazil accounts for
60%) followed by South Asia (where India accounts for 68%). Annual production of dairy cattle
manure was largest in South Asia (where India accounts for 68%). The largest annual manure
production by buffaloes occurs in East Asia (China accounts for 99%) and South Asia (India
accounts for 70%). The largest quantity of broiler chicken manure was recorded in South-Eastern
Asia, where Indonesia accounts for 76% of broiler chicken manure in that region. The next largest



production was recorded in South Asia where Pakistan and Iran account for 42% and 37% of the
regional production (Table 7).
**Table 7**. Estimated total amount of manure potentially produced annually (in 1000 tonnes on dry matter
basis) across different regions. All values were estimated using FAOSTAT data (see methods).

| Continent | Region | Non-dairy | Dairy | Buffalo | Pigs | Broilers | Layers | Ducks | Horses |
|---|---|---|---|---|---|---|---|---|---|
| Africa | Eastern Africa | 240031 | 120362 | 0 | 2550 | 10222 | 750 | 528 | 2869 |
| | Middle Africa | 80722 | 6911 | 0 | 1109 | 4451 | 91 | 5 | 1762 |
| | Northern Africa | 52238 | 40711 | 3980 | 5 | 19224 | 1184 | 472 | 1667 |
| | Southern Africa | 29259 | 4949 | 0 | 221 | 5205 | 289 | 26 | 591 |
| | Western Africa | 121460 | 34941 | 0 | 2116 | 15113 | 1717 | 87 | 2860 |
| Americas | Caribbean | 14020 | 3780 | 11 | 550 | 9762 | 181 | 18 | 2451 |
| | Central America | 81531 | 16973 | 0 | 3233 | 18461 | 1818 | 534 | 10010 |
| | Northern America | 172586 | 32242 | 0 | 12609 | 62529 | 3140 | 567 | 14893 |
| | South America | 598417 | 84428 | 3450 | 9312 | 79881 | 3305 | 580 | 17176 |
| Asia | Central Asia | 28592 | 32850 | 47 | 126 | 2799 | 476 | 5 | 5140 |
| | Eastern Asia | 122616 | 25012 | 49716 | 56895 | 89489 | 23943 | 45644 | 10647 |
| | South-Eastern Asia | 87968 | 15938 | 24457 | 11164 | 165840 | 5127 | 13569 | 1242 |
| | Southern Asia | 369829 | 242073 | 286745 | 1478 | 102379 | 5349 | 6009 | 1451 |
| | Western Asia | 29132 | 32374 | 1092 | 125 | 27461 | 1734 | 47 | 453 |
| Europe | Eastern Europe | 41461 | 45218 | 91 | 7467 | 26523 | 2856 | 2885 | 3192 |
| | Northern Europe | 31161 | 16010 | 0 | 3350 | 5871 | 634 | 3853 | 1014 |
| | Southern Europe | 24594 | 13820 | 746 | 6688 | 2229 | 242 | 17 | 162 |
| | Western Europe | 52432 | 33624 | 19 | 8601 | 14392 | 1234 | 1467 | 578 |
| Oceania | Australia and NewZ | 52801 | 19917 | 0 | 365 | 4024 | 147 | 86 | 402 |
| | Melanesia | 791 | 106 | 0 | 336 | 335 | 19 | 8 | 92 |
| | Micronesia | 28 | 9 | 0 | 7 | 26 | 2 | 0 | 0 |
| | Polynesia | 138 | 10 | 0 | 37 | 35 | 2 | 2 | 22 |
| | Total | **2231803** | **822253** | **370355** | **128344** | **666246** | **54234** | **76408** | **78672** |



## 3.4. Wood residues

Globally, an estimated 0.23 billion tonnes of wood residues are produced every year (Table 8), but
the largest production occurs in East Asia (China producing the highest) followed by South
America and North America where Brazil and USA have the highest production, respectively.
Annual wood residue production was highest in China (95.1 million tonnes) followed by Brazil
(18.8 million tonnes). The values presented in Table 8 are based on countries for which data were
available in FAOSTAT. Since data are not available for all countries in many regions, it was not
possible to calculate the residue production per country as a proportion of the total production in the
respective region. Countries in the Caribbean, Central Asia, Middle Africa, Western Africa,
Northern Africa and Southern Asia are poorly represented (Table 8).






**Table 8**. Estimated total annual wood residue potentially produced (in 1000 tonnes on dry matter basis)
across different regions. All values were estimated using FAOSTAT data (see methods).

|  | Region | Wood residues | Countries where data are available |
|---|---|---|---|
| Africa | Eastern Africa | 112 | Ethiopia, Kenya, Malawi, Madagascar, Mauritius, Zambia |
|  | Middle Africa | 15.7 | Cameroon |
|  | Northern Africa | 119.1 | Sudan, Tunisia |
|  | Western Africa | 609.4 | Mali, Cote d'Ivore |
|  | Southern Africa | 514.5 | South Africa |
| Americas | Caribbean | 0.6 | Cuba |
|  | Central America | 1044.5 | Costa Rica, Guatemala, Honduras, Nicaragua, Panama |
|  | Northern America | 22610.3 | Canada, USA |
|  | South America | 24798.8 | Argentina, Bolivia, Brazil, Chile, Colombia, Ecuador, Suriname, Venezuela, Uruguay |
| Asia | Central Asia | 1.5 | Kazakhstan, Kirghizstan |
|  | Eastern Asia | 101867.0 | China, South Korea, Japan |
|  | Southern Asia | 3.3 | Bhutan, Sri Lanka |
|  | South-Eastern Asia | 8815.2 | Indonesia, Malaysia, Viet Nam |
|  | Western Asia | 966.8 | Azerbaijan, Cyprus, Georgia, Israel, Turkey |
| Europe | Eastern Europe | 19810.6 | Belarus, Bulgaria, Czechia, Hungary, Moldova, Poland, Romania, Russia, Slovakia, Ukraine |
|  | Northern Europe | 19428.2 | Estonia, Finland, Ireland, Latvia, Lithuania, Norway, Sweden, United Kingdom |
|  | Southern Europe | 4412.3 | Albania, Bosnia, Croatia, Greece, Montenegro, Portugal, Serbia, Slovenia, Spain |
|  | Western Europe | 18207.5 | Austria, Belgium, France, Germany, Luxembourg, The Netherlands |
| Oceania | Australia and New Zealand | 2535.8 | Australia |
|  | **Total** | **225873** |  |


## 4. Discussion

The preceding sections have presented preliminary estimates of the quantities of agricultural
residues and by-products for selected crops and animals available in the OMD. Due to the lack of
uniform methodology and data across countries, it was not possible to estimate the quantities of
residues produced by all crops and agro-processing activities. Nevertheless, OMD is a living tool
that will be updated and enriched as data become available to build a solid reference resource for
industry, researchers and decision-makers in soil health management, pollution risk reduction,
bioenergy production and other sectors. The OMD is envisaged to complement existing databases
such as FAOSTAT, FishStat and organic resource quality databases such as Phyllis.  The residue
estimates in OMD may be used for various purposes including estimation of availability for soil
amendment, animal feed, bioenergy and other agricultural activities such as mushroom production.
The use of agricultural and forestry residues and by-products for soil amendment may be



constrained by these competing uses (Duncan et al., 2016; Ji et al., 2018). The following sections
will discuss the production and competing uses of agricultural, fisheries and forestry residues, and
the opportunities and challenges for their use as soil amendment.

**4.1. Crop residues**
The estimates provided for the selected crops (Table 3) reveal that large quantities of crop residue
biomass are produced annually. The estimated total annual crop residue produced by the top cereal
and legume crops across the different regions indicate the high potential for their use in soil
amendment and contribution to bioeconomy processes. Depending on the availability of technology
for recovery, some of the crop residues produced may be used for recycling in bioenergy production
and use as soil amendments. Raw crop residues such as straw can be incorporated into the soil or
applied on the soil surface as a mulch, and this can reduce erosion, maintain soil moisture and add
carbon and nutrients to the soil. A growing body of meta-analyses have provided compelling
evidence that residue retention significantly increases crop yields, soil nutrient stocks, water use
efficiency, carbon sequestration, microbial diversity and functionality (Shu et al., 2022; Wang et al.,
2020). Significant increases in soil organic carbon (SOC) have been achieved following residue
retention relative to inorganic fertilization under residue removal (Wang et al., 2020). This is
because soil incorporation of residues provides a direct carbon source for SOC formation. In a
global meta-analysis of 219 studies, Shu et al. (2022) showed significant improvement in microbial
diversity, richness and community structure (by >100%) following application of crop residues
compared to mineral fertilization.

While crop residues can contribute to enhancing soil organic carbon stocks and nutrient

availability to crops, and reduce soil erosion, not all crop residues produced are readily available as
a soil amendment. Some of the crop residue is burnt in the field or used as fuel for domestic
purposes, for animal feed and/or bedding, mushroom production, construction, industrial
applications (FAO, 2022a; Ji et al., 2018). In some cropping systems and regions, residues are



burned in the field during land preparation because it is the easiest option for farmers. For example,
the intensification of rice cropping with high-yielding and short-duration varieties in Asia has
resulted in larger volumes of rice straw, which must be managed over a very short time between
two or three cropping rounds per year (Van Hung et al., 2020). In such systems, soil application of
residue poses challenges due to the insufficient time for decomposition of the straw, which hinders
crop establishment. This has led to an increase in open field burning of rice straw in some Asian
countries (Lin and Begho, 2022; Van Hung et al., 2020).

In some regions of the world farmers remove residues to feed animals or use them as

beddings. In the EU countries, around 28 million tonnes of crop residues are used for animal
bedding annually (Scarlat et al., 2010). About 16% of the collectible crop residues is used as animal
bedding in Europe (Monforti et al., 2013). Crop residues are also used as fuel in industrial and
domestic set-ups. For example, in rural areas in Africa and Asia, crop residues are used for cooking.
There is also a growing interest in the use of crop residues for the generation of biofuels as
alternatives to fossil fuels and industrial applications including textiles, natural fibres, polymers,
biosorbents and reinforcement material in composites (Siqueira et al., 2022).

Of the residues produced annually, only a small fraction may be recovered because the

collection, storage and transportation of raw residues poses challenges for their use outside their
production area. One way to reduce the cost of transport and increase their use is to convert bulky
residues and by-products into briquettes, pellets, biochar or anaerobic digestate that can be more
easily handled and transported than the raw residues (Bora et al., 2020). In some regions, the short
time frame between two cropping seasons may not allow collection of the available residues (FAO,
2014; 2021). Even when collection is feasible, the cost of transportation may limit soil application
far from the farm where the residues were produced. This may be overcome by mechanized
collection, high-density compaction, briquetting, pelletizing or on-site processing (e.g., composting
or anaerobic digestion). High-density compaction can reduce the volume of crop residues thus
making it easier to store and transport over a long distance. For example, the volumetric weight of
mechanically compacted rice straw bales is 50–100% higher than that of loose straw. Briquetting
and pelletizing can further increase the volumetric weight of baled straw by up to 700% and reduce
transportation costs by more than 60% (Balingbing et al., 2020).

The quality of residues may play a critical role in the build-up of carbon and nutrients in the

soil (Cotrufo et al., 2013) against the backdrop of the importance of the soil ecosystem (Schmidt et
al., 2011). The carbon content of residues is about 30-50% (Table 1). The nitrogen content of
various cereal straws ranges between 0.3 and 2.8%, and only pulse straws are relatively nitrogen-
rich (Table 1). With low C:N ratios (Table 1), residues from legumes are likely to decompose more
rapidly than cereals. The phosphorus and potassium content of most residues is 0.05-0.3% and 0.2-
2%, respectively (Table 1). As such, crop residues represent a substantial store of carbon and
nutrients that can be used as inputs for soil amendment. A role of crop residue incorporation that
has remained less appreciated is their contribution to soil micronutrient stocks especially sulphur,
calcium, magnesium, zinc and silicon that are often not part of the recommended fertilizers. Where
straw is incorporated, reserves of soil nitrogen, phosphorus, potassium and silicon have also known
to be maintained at acceptable levels (Dobermann and Fairhurst, 2002).

### 4.2. Agro-processing by-products

**4.2. Agro-processing by-products**
Our estimates indicate that substantial quantities of by-products are produced every year, but with a
great deal of variability across regions. Unlike crop residues, most of the by-products are produced
in localized processing plants, which makes their collection more convenient. However, some of the
by-products may not be available for soil amendment as they have various uses. For example, husks
of rice and sugarcane bagasse are mostly used as fuel in the rice and sugar mills. Rice husk is also
used as an insulating material. In crops such as oil palm, cocoa and coffee, the processing also
occurs in a few countries where the commodities are grown on commercial scale. Although oil palm
is widely cultivated in plantations across the humid tropics of Asia, Africa and the Americas, over
90% of the global palm oil production occurs in just five countries, namely, Indonesia (58.8%),





Malaysia (25.6%), Thailand (3.9%), Colombia (2.9%) and Nigeria (1.4%) (Murph et al., 2021).
Although the oil palm industry is one of the best sources of organic inputs for agricultural use (Adu
et al., 2022; Embrandiri et al., 2012), the residues may not be available for direct soil application in
areas far from processing plants. However, this can be circumvented through conversion into
compost or digestates, which are easier to handle and transport.

As with crop residues, there are challenges to the availability of by-products from fish

processing for soil application. Some fish parts, especially viscera, deteriorate very rapidly and
therefore they require preserving as soon as possible after being produced. This is not always
possible due to inadequate processing facilities or limited volumes making recovery of the by-
products unprofitable. When fish are processed to fillets at sea, viscera, the head and frames are
often discarded since refrigeration facilities are used for the most valuable product (Olsen et al.,

2014).


**4.3. Livestock manure**

Our estimates in Table 7 show that large quantities of manure are produced annually albeit large
variability across regions. These estimates include both manure management systems and manure
left on pasture. Only a fifth of livestock manure produced is returned to soil due to various
constraints. For example, much of the manure produced may not be available for application as soil
amendment because over 70% is directly deposited on pasture (FAO, 2018). Manure applied to soil
can be a significant source of macronutrients and micronutrients (FAO, 2018; Sileshi et al., 2019).
In addition, manure is a significant source of organic matter, which is a key determinant of soil
health (FAO, 2018). For example, globally manure applied to soil was estimated to contribute 24
and 31 million tonnes of nitrogen per annum based on IPCC Tier 1 and Tier 2 approaches,
respectively (FAO, 2018). According to van Dijk et al. (2016), manure application on soil
constitutes approximately 53% of the P and 33% of the N applied annually to agricultural land in
the EU27.



Even if manure is available in abundance, its application may be constrained by

environmental quality and economic considerations in some jurisdictions. For example, in the USA,
the Environmental Protection Agency regulation requires large animal feeding operations to meet
nutrient planning requirements for land application of manure. Similarly, according to the EU
Council Directive 91/676/EEC, the amount of livestock manure applied to land each year shall not
exceed 170 kg N per hectare. Legislation may also forbid manure application during certain periods
(e.g., in non-cropping seasons) or land that would otherwise lead to environmental impact through
run off or nutrient leaching (Loyon, 2018).

The bulky nature of manure limits the areas over which it can be economically applied.

According to Paudel et al. (2009), the economically optimal distances for dairy manure application
is 30 km for nitrogen and 15 km each for phosphorus and potassium to meet the recommended N,
$P_2O_5$ and $K_2O$ needs on cropland. Conversion of manure into anaerobic digestate or compost can
circumvent the handling, storage and transportation costs of raw manure from intensive animal
production units. When efficiently managed and recycled within agricultural systems, livestock
manure represents a large source of plant nutrients that can reduce the need for synthetic fertilizer
inputs and reduce GHG emissions (FAO, 2018). Manure may be applied by injection, band
application, surface spreading or incorporation (Emmerling et al., 2020). Injection has been cited as
the best application method to reduce $NH_3$ emissions, while surface application using splash plates
has been banned in most European countries because of its strong impact on $NH_3$ emission
(Emmerling et al., 2020).

## 672   4.4. Wood residues

Wood residues are obviously underestimated for many regions because data were unavailable for
some countries. Among the countries for which data exist, annual wood residue production was
highest in China and Brazil, representing 42% and 8.3% of the annual global wood residue
production. Wood log production in Brazil generates about 50.8 million $m^3$ of lignocellulosic





residue yearly (Domingues et al., 2017). Assuming a wood density of ~450 kg m$^3$ this value is
approximately 22.9 million tonnes, which is slightly higher than 18.8 million tonnes in our
database. The competing uses of wood residues include use as woodfuel for domestic purposes
(Flammini et al., 2022), bioenergy generation (Karan and Hamelin, 2020) and as raw materials for
the manufacture of agglomerated products such as pulp, particle board and fibreboard (FAO,
2022f). Although wood residues could be potentially used for soil amendment after processing (e.g.,
wood-ash, biochar, compost, etc.), the proportion actually available may be small due to their
various competing uses. Agroforestry trees and plantation crops such as coconut, oil palms, and
rubber generate considerable amounts of woody and leafy biomass from pruning and lopping. A
large proportion of such residues can be used for soil amendment directly or after processing into
compost or biochar (Bluhm and Lehmann, 2023). However, data were not readily available for
these residues, and therefore it was not possible to collate their quantities in the OMD.

## 5. Limitations of the OMD and challenges ahead

One of the key limitations of the OMD is our inability to provide global estimate of all residues
from agriculture, fisheries and forestry. There are also uncertainties associated with the estimates
presented. The effort to compile estimates of quantities of residues and by-products was hampered
by the lack of methods for conversion of primary products to residues and industry standards for
collection and aggregation of such data. For example, we did not included the quantities of residues
produced by minor crops, fruit trees and other trees in agroforestry and forestry. The OMD also
does not contain the quantities of by-products such as biochar, compost and digestate produced due
to lack of data and reporting frameworks on their production. By-products of secondary processing
that occurs in the breweries and beverage industry could also not be compiled due to lack of data.
By-products from capture fisheries were estimated only for a few species because conversion
factors were unavailable for the majority of species. Even for those species where conversion
factors were available, residues from capture fisheries were probably underestimated by a large



margin because recovery of inedible parts is challenging. This is because the fish are processed at
sea, and non-edible parts may be discarded in the sea (Olsen et al., 2014). Commercial fish products
are often directly processed on-board vessels and, by the time they are landed, the fish have been
frozen, gutted, headed, and/or processed, leading to a considerable change from their original
weight. This leaves a great deal of uncertainty about estimation of fisheries by-products.

This work only provides an inventory of the various residues at the country level, which is

valuable in its own right. However, further work needs to be done to produce a global map of
carbon and nutrients from residues at much greater spatial distribution and finer resolution than
individual countries to inform policy and good practice for more efficient allocation of biomass
resources. This requires further work and deemed outside the scope of this publication.

Due to lack of basic data, this work was unable to determine the proportion of the residues in

each category that is actually available for use as soil amendment. Even where data were available,
legislative and regulatory issues may limit their use as soil amendments. For example,
environmental concerns of pollution by antibiotics, heavy metals and pathogens have led to
regulations on direct spread of manure on land (Font-Palma, 2019). Strict regulations such as those
under the EU Nitrates Directive 91/676/EEC (EEC, 1991) mean that only a small proportion of the
total volume of manure produced can be used for soil amendment. It is also forbidden to apply
manure or anaerobic digestate at particular times in the year or on certain types of land (Loyon,
2018). In some jurisdictions, organic matter that has been designated as waste may be subject to
regulatory restrictions on how it can subsequently be used or managed (Loyon, 2018). In this
analysis, it was not possible to evaluate the extent to which national policies and regulatory
frameworks governing the classification of organic matter streams as wastes or by-products, and
waste management can provide incentives or not to the use of organic inputs for soil amendment.
Legislation banning residue burning and incentives for farmers to adopt good agricultural practices
can also incentivise appropriate use of agricultural residues. For example, EU Regulation No
1307/2013 has established rules for direct payments to farmers under support schemes within the





framework of the common agricultural policy. To receive full payments, farmers in the member
states have to comply with statutory management requirements and standards for good agricultural
and environmental conditions, and the requirements of 'greening' (Heyl et al., 2021). Quantitative
targets are used to incentivize the implementation of agricultural practices that increase SOC stocks
(Bruni et al., 2022). For example, the EU Mission Board for Soil Health and Food proposed a series
of quantitative targets for soils to become healthier. Among them, the current SOC losses of about
0.5% per year in the 20 cm soil depth of croplands should be reversed to an increase of 0.1–0.4%
per year by 2030 (Bruni et al., 2022). Such targets and related regulations will have implications for
how and where agricultural residues can be used for soil amendment.

Transport costs may also hinder the use of the excess volume produced in one region in other

regions. In some regions, anaerobic digestate is produced in excess of its agricultural assimilation
potential (Torrijos, 2016). For example, in the EU digestate production reached 56 million tonnes
per annum by 2010, of which 80% could be recycled back into agriculture (Kizito et al., 2019).
Similarly, in China the annual digestate production is approximately 2.3 billion tonnes of which less
than 70% is recycled back to agriculture due to land limitations (Kizito et al., 2019). These
observations highlight the need to explore opportunities for use of residues and by-products outside
the country where they are produced.

**Data availability:** The OMD data is available at: https://doi.org/10.5281/zenodo.8158727
(Sileshi et al., 2023).

**6. Conclusions**
This work has provided typologies, definitions and quantities of the various agricultural residues
and by-products, which can be useful for the inventory and estimation of the various residue
streams potentially available for recycling in agriculture, bioenergy and other sectors. The OMD is
the first of its kind to consolidate biomass estimates of residues and by-products from agriculture,





fisheries, forestry and allied industries globally. The OMD will be continuously updated as new
production data are published in FAOSTAT and will be publicly available for use by different
decision-makers. It is hoped to contribute to the Better Production and Better Environment
dimensions of FAO's Strategic Framework 2022-2031 supporting the 2030 Agenda. The OMD and
associated products are also expected to contribute to evidence-based policies and actions in support
of the transition towards a circular economy, and more sustainable agriculture and food systems.
Currently, the estimates in OMD are available only at the national level. Therefore, finer scale data
are urgently needed for spatial targeting of residues and by-products for various applications.
Detailed site-specific inventory of various categories of residues and their local uses are highly
recommended.

## Authors' contributions

EB, GWS, FNT conceptualized and designed the study. GWS, JL developed the methodology and
GWS conducted data curation and formal analysis. GWS, EB wrote and edited the manuscript,
while JL, FNT reviewed and edited the manuscript. EB funding acquisition. All authors have read
and approved the final version of the manuscript.

## Competing interests

One author (FNT) is a Topical Editor of *Earth Systems Science Data*.

## Disclaimer

The views expressed in this paper are the authors' only and do not necessarily reflect those of FAO.

## Acknowledgements

FAOSTAT is supported by FAO's member countries. We acknowledge the efforts of national
experts who provide the statistics on food and agriculture, as well as statistics on energy use, that
are the basis of this effort. This work was financially supported by the McKnight Foundation Grant
#15-113 "Strengthening Multistakeholder Cooperation on Agroecological Approaches for
Sustainable Agriculture".



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
