# Peer review of "agriculture, fisheries, forestry and related industries"

_Earth System Science Data, 2023_

## Author Response (AR1)

Dear Editor,
On behalf of my co-authors, I would like to thank you for giving us the opportunity to submit a revised version of our manuscript. We have now accepted all the comments and made the revision following the comments and suggestions provided by both reviewers. We have outlined below how and where we have made the recommended corrections.

Kind regards
Sileshi G Weldesemayat
Edmundo Barrios

**Reviewer #1**

Review of "**Organic Matter Database (OMD): Consolidating global residue data from agriculture, fisheries, forestry and related industries**" by Sileshi et al.

**ESSD-2023-288**

The authors carried out an interesting topic aiming at unifying the definitions, typologies, and methods of residues and estimating global organic matter database of residues. However, despite its valuable goals, several critical limitations significantly curtail its application and value. The oversimplified methodology, overbroad definition, and unclear use of residues pose considerable constraints. The current version appears more suited as a FAO report or a methodological document. Further improvements are essential before considering it for publication in ESSD, as detailed below.

We would like to thank the reviewer for the insightful comments and suggestions.

1. **Ambiguity in objectives:** This paper focuses on estimating a global organic matter database of residues from diverse production systems. However, it predominantly emphasizes soil application of residues as the effective way, with recurrent mentions of soil fertility, soil health, and soil amendment. While this approach might suit some residues, the recommendation to uniformly transfer all residues for soil application neglects their multifaceted existing uses. Lots of these residues have already been used in a variety of effective ways, such as feed and bioenergy. I couldn't agree with the recommendation for making use of all kinds of residues for soil application without considering the practical efficiency.

**Response**: We agree with the reviewer that many of these residues are already being used in a variety of ways. Our intention was not to recommend the use of all kinds of residues for soil application. Although a large body of literature exists on the value of agricultural residues as soil amendments, and in maintaining soil carbon, their use has often been neglected in policy circles. Information is also not systematically collected on the production and use of agricultural residues as soil amendments at the national or global level. Therefore, this work focused on the production and use of agricultural residues to raise awareness about the dangers of current practices (e.g., burning, disposal into water bodies) and the potential use, when feasible, of agricultural residues for soil amendment. We have now highlighted this better in the introductory section (lines 45-55) in the revised manuscript. We have also clarified the objectives. We have also highlighted the other alternative uses of the various categories of residues when revising the manuscript.

**2. Limitations in Database Applicability**

- **Oversimplified methodology**. The methodology applied seems overly simplistic and general, overlooking the difference between countries and over time. For example, the total agro-processing by-product of barley is calculated by multiplying barley product with 0.29 for all countries and all years. Another example is manure production, which varies substantially due to many factors such as species, feed, and environment. Using average manure excretion rates introduces substantial biases. I would suggest using regional and country-specific parameters, at least for the major producers, and publishing these parameters.

**Response**: We totally agree with the reviewer that in some cases (but not all) the methodology we used does not account for difference between countries and changes over time. For example, for agro-processing by-products such as those from barley we did not have information on country level residue product ratios or other conversion factors. Similarly, regional or country-specific parameters are lacking for manure excretion rates. The methodology we applied has been the commonly used approach in scientific publications (e.g., Bedoić et al., 2019; Bentsen et al., 2014; Hamelin et al., 2019; Karan et al., 2021; Ronzon and Piotrowski, 2017 now newly added to our manuscript in response to the suggestion). In the revised manuscript, we have highlighted this as a limitation of this work, and we have now recommended generation of data to come up with regional or country-specific parameters for the major residues.

- **Overbroad definition**. The residue defined in this paper is too broad for different products. For example, the residue of meat processing by-products is simply defined as the rest part of an animal by excluding carcasses and hides/skins. However, different parts of the residue such as head, feet, fat, and blood can be used in a variety of ways. I would suggest further splitting the residues into finer categories for more accurate estimations and insights into their varied utilization.

**Response**: We totally agree with the reviewer regarding the broad definitions, and that different parts of the animal such as head, feet, fat, and blood can be used in a variety of ways. We have now highlighted this in the revised discussion section (lines 660-663). We have not included edible parts in the calculations. We had further split the residues into

finer categories and made the calculations specifically focussing on parts that are often disposed off in slaughter houses. Since data are not available from slaughter houses on specific uses of those disposed of, we were unable to establish their alternative uses and considered them as residues. In the revised manuscript, we have now explained this.

- **Unclear use of residues**. Residues from different sectors can be used differently. Some of the residues have already been effectively utilized as fertilizer, biogas, feed, etc. The paper overlooks the diverse and existing effective uses of residues. I think it would be more valuable to provide the estimates on those unused or wasted residues. I would suggest estimating the share of different uses of each residue.

**Response**: We totally agree with the reviewer regarding lack of clarity on the use of residues. This is due to lack of information on the alternative uses of residues and absence of data on the quantities used for the different purposes at the country level. Data is simply lacking on the diverse and existing effective uses of residues at the national level. Even at the global level, empirical data are virtually lacking on the share of different uses of each residue. The only information available so far is global estimates provided by Smerald and colleagues published as Scientific Data in 2023. This information is also available only for crop residues. The process to estimate alternative uses is so complicated that it is not realistically possible at this stage for us to provide the estimates on those unused or wasted residues. In the revised manuscript we have highlighted this data gap and provided recommendations for future research (line 752-754).

**3. Inclusion of residue quality:** I would suggest adding the quality of the residues such as carbon and nutrients content along with quantity. Such data would add depth and clarity to readers. It is difficult for readers to understand the values of tonnes of fishery residues. It is also difficult to compare the values of residues among different products by using quantity.

**Response**: We have now added residue quality as a supplementary material. Following is the DOI associated with it in ZENODO. 10.5281/zenodo.10450921

4. I would suggest re-organizing the overall structure of this paper to make different sections to be closer connected and easier to follow.

- **Introduction:** I would suggest introducing other uses of residues. The current content emphasizes too much on the importance of residues in soil amendments via crop residue burning, synthetic fertilizer use, and a combination of organic and inorganic fertilizers. Without introducing other uses, readers may think the goal of this paper in utilizing residues is for soil fertility. The characteristics of residues such as lignin and polyphenol contents discussed in line 91-108 seems irrelevant to this section. I would suggest removing them or moving them to discussion.

**Response**: We agree with the reviewer. In the initial submission we had described the alternative uses in the discussion section under each residue type. In the revised manuscript

we have describe other uses of residues in the Introduction section (line 70-75). We have now removed the sentence referring to lignin and polyphenol contents (earlier line 91-108.

- **Methods:** I couldn't find any results and supplementary database of carbon and nutrients of residues described in lines 147-158 and Table 1.

  **Response**: We have now added a file on residue quality as a supplementary material. We have already uploaded a file for this supplementary material on ZENODO with a DO: 10.5281/zenodo.10450921

- **Results:** The oversimplified methodology makes the comparison of residues between countries meaningless. Since the uniform parameters were applied to all countries, the difference in residue reflects the differences in production among countries.

  Response: In the revised manuscript we now use regional or country-specific parameters for the major producers wherever such parameters are available.

- **Discussion:** I feel the discussion section is disconnected from the previous context. I would suggest adding some discussions about the different shares of residues by countries and regions, as well as the difference of uses in residues across regions and countries. For example, comparing maize and soybean residues against sugar by-products in North America could add depth.

Response: In the revised manuscript we have added text to discussion on the different shares of residues by countries and regions, as well as the difference in uses of residues across regions and countries. For example, in line 504-508 and Table 5 we have presented sugarcane bagasse production. We have also discussed the competing uses of bagasse in line 699-706. We believe comparison of maize and soybean residues against sugar by-products in North America is not straightforward due to the limited amount of data on availability of bagasse for other uses. Unlike crop residues, much of the bagasse produced is used for steam generation in sugar mills and whatever remains is burnt because dry bagasse is known to be a fire hazard.

**Reviewer #2**

Dear authors,

Thanks for the initiative to develop and share an organic matter database. Great effort and most useful!

To make the data shared more meaningful, the following issues require attention: (1) the dataset with nutrient concentrations is lacking; (2) the Excel files are in need of readme sheets with definitions, meaning of abbreviations, meaning of column headings, data sources and equations used; (3) no information is shared on uncertainties or ranges of the estimates.

**Response**: We would like to thank the reviewer for these suggestions. We have now (1) uploaded dataset with nutrient concentrations; (2) uploaded a readme file with the definitions of all terms and data sources; (3) provided the 95% confidence limits as a measure of uncertainty around the global total residue production for each crop. We were unable to accommodate uncertainty estimates for each region as the tables will be cluttered and become too complex.

More detailed comments below.

**Specific comments**

- You state that the estimates are 'preliminary' and need to be used with caution. Can you give indications/levels of uncertainty or ranges for the different estimates you calculate?

**Response**: We have now provided the 95% confidence limit (CLs) as a measure of uncertainty around the global total residue production for each crop. We have also indicated the CLs in the text where reference was made to the values. We are unable to provide uncertainty estimates for each region as the tables will be cluttered.

- Please add readme sheets with definitions, meaning of abbreviations, meaning of column headings, data sources and equations used to all Excel files.

**Response**: We have now uploaded a ReadMe file with the definitions of all terms and data sources.

- The data file with nutrient concentrations is missing. Please add.

**Response**: We have now uploaded the datafile containing nutrient concentrations

- Crop residues and by-products can be used for multiple purposes (feed, fuel, material, soil amendment), the optimal usage will most likely depend on the environmental or economic objective and context. Yet the article gives focuses mainly on the utilization as soil amendment, the reasoning being unclear. Can you give more balanced attention to the different potential purposes of crop residues and by-products?

- The organic resource database developed by Palm et al. (2001) is mentioned in Line 96, but not included in Lines 149-154. Is the data from Palm et al. (2001) included in the OMD?

**Response**: We have now added a file on residue quality as a supplementary material including data from Palm et al. (2001). We have already uploaded a file for this supplementary material on ZENODO with a DO: 10.5281/zenodo.10450921

- Line 755: "*The OMD will be continuously updated as new production data are published in FAOSTAT*" and Lines 539-542: "*..OMD is a living tool that will be updated and enriched as data become available to build a solid reference resource.. *" -> How will this continuous update look like in practice? Can you share more details on the logistics of this effort? Is this automated? is there a timeframe or an ongoing project?

**Response**: Currently, the process is not automated. However, in the coming years we plan to automate the updating of the OMD by linking it to FAOSTAT.

- On the estimated quantities of by-products (Section 3.2): What happens when products are processed in a different country than their origin? (such as soybean or cocoa processing in the Netherlands)? Do you account for this when calculating available quantities of by-products?

**Response**: We were unable to account for this when calculating available quantities of by-products. This would require additional data on export and import. We have indicated this as one of the limitations of this work and as a future endeavour in the development new versions of the OMD.

- Line 643: "*Only a fifth of livestock manure produced is returned to soil due to various constraints. For example, much of the manure produced may not be available for application as soil amendment because over 70% is directly deposited on pasture (FAO, 2018)*" -> Please revise the first part of this statement. When manure is directly deposited on pasture it is also returned to the soil.

**Response**: We totally agree with the reviewer. Here we were meaning soil application on cropland. In the revised manuscript we have corrected that sentence.

- Any insights on current usage of by-products and residues would be useful as this would show what 'additional' or 'alternative' utilization options would be compared to the baseline.

**Response**: In the revised manuscript we have described the current usage under each by-product and residue in the discussion section. For example, in the discussion under crop residues we have extensively described the additional or alternative uses (lines 5091-608). Similarly, under agro-processing we have provided the additional or alternative uses in line 678-680, lines 371-734 under wood residues.

**Technical corrections**

- Line 179: "*By-products may be products of either primary or secondary processing of crops*"
-> also from animal product processing?

**Response**: We have now corrected this

- Line 271: has been observed to control

**Response**: We have now amended this sentence as suggested

- Line 287: Slaughterhouse by-products consist of...

**Response**: We have now amended this sentence as suggested

- Excel file Coffee cocoa and oilpalm: Please check heading and unit of column J

**Response**: We have now checked the heading in column. It is correct. It represents "Production expressed in tonnes.

- Excel file cop residues: What is meant with 'production'? What is the definition and is this per year?

**Response**: Production refers to the total quantity of residues produced per country in a given year. We have now provided that definition in the text (line 166-167) and the readme file.

---

## Author Response (AR2)

Dear Editor,

On behalf of my co-authors, we would like to thank you for giving us the opportunity to submit a revised version of our manuscript. We have now accepted the comments and made the required revision following the comments and suggestions provided by the reviewers. We would like to highlight that most of our original estimates were produced using country-specific conversion factors, and the concerns by the reviewer were partly addressed. This was probably not clear in the manuscript, and we have now indicated that our estimates were country-specific by inserting the qualifier "country-specific" wherever applicable. Below are our specific responses. We have also highlighted the changes we have made in the manuscript in track changes mode.

Sincerely yours

Sileshi G. Weldesemayat and Edmundo Barrios

**Editors comments**

Thanks for the authors' and reviewers' efforts in improving this manuscript. We noticed the conflicting reviews from two referees, emphasizing the need to account for cross-country and temporal variations in conversion factors and residue fate. Could the authors consider the following improvements: (1) provide an estimation range by synthesizing and comparing region-specific and globally uniform conversion coefficients, discussing how limited knowledge of these coefficients may affect the database, and (2) discuss residue fate and changes based on available information from specific countries or regions? While this may not produce a more detailed global database, it could inform data users about current cross-country status and limitations in the global database, with quantitative insights.

Response: We would like to thank the Editor for offering us the opportunity. Our estimates for agro-processing byproducts of crops, poultry, meat, fisheries and forestry by-products were based on country-specific conversion coefficients. This was probably not clear in the manuscript, and we have now indicated that by inserting the qualifier "country-specific" wherever applicable. Unfortunately, we were unable to produce similar estimates for crop residues and manure, comparing region-specific and globally uniform conversion coefficients because of lack of country-specific or region-specific conversion coefficients. In lines 622-625 we have now discussed how limited knowledge of country-specific conversion coefficients may affect the database. We have also indicated this in the conclusion section (line 862-864). We were also not able to disaggregate the total residue into the different categories due to lack of data on the proportion fed to animals, burnt or left on the ground. We have now indicated this in line 627-629, and recommended an inventory of the competing uses and fate of the various residues and wastes in each country to improve data availability in the conclusion section (line 867-868).

**Reviewer 1**

The authors have made significant improvements to the manuscript, which is now well organized and presented. One personal suggestion is to add some figures to show the spatial and temporal patterns and other interesting findings. However, as this is a data paper, I still have concerns about the dataset.

Currently, the data only presents the "total residue" of different productions using a constant conversion factor for all countries over time. In my opinion, it's overly simplistic and less valuable to the community. It may be more suitable for methodology documentation for FAO rather than a comprehensive dataset.

I suggest the following improvements for future publication: (1) enhance the methodology to differenciate between countries and time periods. (2) disaggregate the total residue into different categories, such as feed, burning, left on ground, etc.

Without these improvements, merely adding more descriptions of the limitations and promising updates in the future do not enhance current dataset's utility and may, in fact, reduce its credibility.

Response: We agree with the reviewer that our database contains some estimates (not all) produced using a constant conversion factor for all countries over time. Note that our estimates for agro-processing byproducts of crops, poultry, meat, fisheries and forestry were based on country-specific conversion coefficients. Analysis differentiating between countries was not possible for crop residues and manure due to lack of approved harvest indices or conversion factors at the country level. We were also not able to disaggregate the total residue into the different categories due to the lack of data on the proportion fed to animals, burnt or left on the ground. In the revised manuscript (line 622-625) we have indicated this as a limitations of our data. At the end of the conclusion, we have now recommended an inventory of the competing uses and fate of the various residues and wastes in each country.